# Regenerative Capacity of Adipose Derived Stem Cells (ADSCs), Comparison with Mesenchymal Stem Cells (MSCs)

**DOI:** 10.3390/ijms20102523

**Published:** 2019-05-22

**Authors:** Loubna Mazini, Luc Rochette, Mohamed Amine, Gabriel Malka

**Affiliations:** 1Laboratoire Cellules Souches et Ingénierie Tissulaire, Centre Interface Applications Médicales CIAM, Université Mohammed VI polytechnique, Ben Guérir 43150, Morocco; gabriel.malka@um6p.ma; 2Equipe d’Accueil (EA 7460), Physiopathologie et Epidémiologie Cérébro-Cardiovasculaires (PEC2), Université de Bourgogne Franche Comté, Faculté des Sciences de Santé, 7 Bd Jeanne d’Arc, 21000 Dijon, France; luc.rochette@u-bourgogne.fr; 3Laboratoire d’Epidémiologie et de Biostatique, Centre Interface Applications Médicales CIAM, Université Mohammed VI polytechnique, Ben Guérir 43150, Morocco; mohamed.amine@um6p.ma; 4Département de Santé Publique et de Médecine Communautaire, Faculté de Médecine et de Pharmacie, Université Cadi Ayyad, Marrakech 40000, Morocco

**Keywords:** mesenchymal stem cells, bone marrow, umbilical cord, adipose tissue, adipose derived stem cells, ADSCs, stem cell therapy, regenerative medicine

## Abstract

Adipose tissue is now on the top one of stem cell sources regarding its accessibility, abundance, and less painful collection procedure when compared to other sources. The adipose derived stem cells (ADSCs) that it contains can be maintained and expanded in culture for long periods of time without losing their differentiation capacity, leading to large cell quantities being increasingly used in cell therapy purposes. Many reports showed that ADSCs-based cell therapy products demonstrated optimal efficacy and efficiency in some clinical indications for both autologous and allogeneic purposes, hence becoming considered as potential tools for replacing, repairing, and regenerating dead or damaged cells. In this review, we analyzed the therapeutic advancement of ADSCs in comparison to bone marrow (BM) and umbilical cord (UC)-mesenchymal stem cells (MSCs) and designed the specific requirements to their best clinical practices and safety. Our analysis was focused on the ADSCs, rather than the whole stromal vascular fraction (SVF) cell populations, to facilitate characterization that is related to their source of origins. Clinical outcomes improvement suggested that these cells hold great promise in stem cell-based therapies in neurodegenerative, cardiovascular, and auto-immunes diseases.

## 1. Introduction

Present within adipose tissue (AT), multipotent mesenchymal/stromal stem cells (MSCs) have been isolated in a reproducible manner and they have changed the thought that the scientific and medical communities have on AT [1]. These cells were isolated in the stromal vascular fraction (SVF) and they were similarly identified to those from bone marrow (BM), are plastic adherent, and typical MSCs, mainly called adipose derived stem cells (ASCs or ADSCs) [1,2]. These ADSCs possess adipogenic, osteogenic, chondrogenic, myogenic, cardiogenic, and neurogenic potential in vitro [3,4], and this plasticity and multipotency have triggered much related research in recent years. ADSCs are considered as tools for replacing, repairing, and regenerating dead or damaged cells. These cells were included in clinical investigations belonging to therapeutic strategies [4,5,6,7,8].

As for hematopoietic stem cells (HSCs), the development of therapeutic strategies presenting more efficacy and safety has become a real challenge in terms of invasive collection and administration, clinical outcomes, and treatment charges. In this fact, BM-, Umbilical Cord (UC)-MSCs, and ADSCs were used as new approaches for stem cell-based therapies in regenerative medicine. The use of enriched ADSCs as cell-Assisted Lipotransfer (CAL) is now largely accepted by plastic surgeons and it was first used to overcome fat necrosis and enhance fat grafting, especially in cosmetic remodeling [9,10,11,12,13]. Currently, pre-clinical and clinical applications of fat, SVF, or enriched ADSCs to treat different diseases are very promising. These investigations have concerned wound defects, vascular ischemia, bone regeneration, neurodegenerative diseases, cartilage tissue defects, cardiovascular injuries, and graft-versus-host disease (GVHD) [10,14,15,16,17,18,19,20,21,22]. These reports have strengthened justifying whether our current knowledge meet the real therapeutic applications of these cells.

Additionally, the widespread clinical use of ADSCs depends on the manner of using them. Being purified or being within their microenvironment is critical for their therapeutic outcomes and might ensure insights regarding the induced side-effects. Another issue is the shift of ADSCs use from autologous to allogeneic setting. The development of biotechnological techniques have improved the use of highly purified ADSCs and newly performed cells [23,24], being proposed for the allogeneic setting in the absence of available autologous cells. In this way, ADSCs might play the primary role in the regenerative medicine of the 21 centuries, provided that risk factors that are related to their manipulation and cryopreservation, their concentration, and route of administration are controlled and standardized.

## 2. Mesenchymal Stem Cells Characteristics

Adult stem cells have been isolated from BM and identified as MSCs [25,26]. Their self-renewal and differentiation abilities raise great interest in cell-based therapy. Alternate sources of these stem cells have been characterized, such as UC tissue, blood, liver, dental pulp, and skin, according to considerations in terms of collection procedure, cell quantity, immaturity, and cell profile [1,27,28,29,30,31]. Moreover, AT and the MSCs that it contains exhibited properties making them more efficient in regenerative medicine [1,30,32,33]. Comparative analysis of these MSCs suggested that, although they share common stem cell properties for MSCs, they markedly differ regarding their population number, proliferation rates and differentiation abilities, and clinical outcomes [34,35,36]. Being defined by the Mesenchymal and Tissue Stem Cell Committee of the International Society for Cellular Therapy (ISCT), MSCs showed marked expression of stromal markers CD73 and CD105, and they were negative for hematopoietic markers CD14, CD34, and CD45 [37]. Additionally, scientific statements that were on the fact that these cells should be plastic adherent, have fibroblast-like morphology and were maintained in culture for long periods. In addition, the tri-lineage potential for osteogenic, adipogenic, and chondrogenic differentiation should be demonstrable. However, cell populations satisfying all of these criteria are likely to be still heterogeneous. Therefore, the term MSCs is currently being used to represent both mesenchymal stem cells and multipotent mesenchymal stromal cells.

The most useful characteristic reported for MSCs is being the non-immunogenic profile. MSCs are widely reported as presenting the Major Histocompatibility Complex (MHC) I, but lacking the MHC II leading to the inactivation of T cells and immunosuppressive properties [38]. Many findings reported an additional immunomodulatory effect by impairing maturation and CD80 and CD86 expression level of dendritic cells, and proliferation and differentiation of human B cells [39,40]. Interestingly, MSCs secrete different chemokines and cytokines and extracellular matrix (ECM) proteins that are involved in various biological process, including hematopoiesis, angiogenesis, leukocyte trafficking, immune, and inflammatory responses [41,42,43]. These evidences hold great promise in MSCs-based therapy allowing for allogeneic transplantation without the need for immunosuppression [44,45,46].

## 3. Types of Stem Cells

There are various types of stem cells that come from different places in the body or they are formed at different times in our lives. Here, this study focused on the major types of stem cells

### 3.1. Bone Marrow-Mesenchymal Stem Cells (BM-MSCs)

BM harvesting is a highly invasive and painful procedure implying general anesthesia and many days for hospital care. BM-MSCs constitute a rare population, with only 0.002% of total stromal cell population [25], and their isolation depend on the patient status and the volume of aspirates. These cells were first identified as accompanying the primary HSCs [26], and they have been reported as multipotent stem cells that are able to differentiate into adipogenic, chondrogenic and are more likely to osteogenic progenitors [47,48]. The most discomforting use of these cells is that their differentiation potential, their number, and maximum life span significantly decrease with age [35,49].

In expansion culture, BM-MSCs stopped growing by passage 11–12, strongly expressed the senescence-related proteins p53, p21, and p16 and they were less clonogenic when compared to those from UC blood (UCB). Their inflammatory properties that were demonstrated in different cell-based therapies were exhibited by the large secretion of Interleukin-1 α (IL-1α), IL-6, IL-8, Tumor Necrosis Factor- α (TNF-α), and Tumor growth Factor-β1 (TGF-β1), but this activity remained lower than the other MSCs sources [5]. Lee et al. have reported 24 genes that were down regulated in BM-MSCs when compared to ADSC, but these genes represent less than 1% of those that were expressed by both cells [50]. BM-MSCs also contributed to tissue repair through their ability to migrate and home to injured sites, by modulating cell surface membrane marker’s expression [51]. 

### 3.2. Umbilical Cord-Mesenchymal Stem Cells (UC-MSCs)

These cells can be isolated from different parts of UC, including Wharton’s Jelly, cord lining, and peri-vascular region. However, almost studies used MSCs that were derived from cord blood and Wharton’s jelly [52,53]. As compared to BM, MSCs frequency was reduced in UCB [31], while their multipotency was maintained for longer periods [54].

UC-MSCs presented significantly lower expression of the senescence markers p53, p21, and p16. They were slightly positive for the embryonic stem cell marker Oct4, Nanog, Sox2, and KLF4, which indicated that these cells are more primitive than the adult stem cells [5]. Interestingly, a recent finding has demonstrated that MSCs that were isolated from male-derived Wharton’s Jelly presented higher gene expression levels of Oct4 and DNA-methyltransferase1 (DNMT1), thus associating stemness profile to gender differences [55]. Their ability to differentiate into the three mesodermic lineages (adipocytes, chondrocytes, and osteocytes) was debated [31,56]; however, the greatest proliferation rates and clonogenic activity, as well as higher chondrogenic differentiation, have been shown [54,57]. As their counterparts, the UC-MSCs were known to secrete growth factors, cytokines, and chemokines, improving different cell repair mechanisms [5]. Nevertheless, their expanded use is specifically due to their low expression of MHC class II antigens, inhibition of T cell, B cell, and NK cell proliferation, and the inactivation of monocytes and dendritic cells [36,58]. Transplanted into animal model, these cells were also able to differentiate into spinal cord tissue with an increasing number of GAP43+ fibers and a higher amount of spared gray in a dose-dependent and repeated manner [59]. Parkinson’s disease, myocardial infarction, and diabetes have also benefited from the differentiation ability of these cells [60,61,62].

### 3.3. Adipose Derived Stem Cells (ADSCs)

Different terminologies were proposed for these cells in the literature: processed lipoaspirates, adipose-derived adult stem cells, adipose mesenchymal stem cells, adipose-derived stromal/stem cells, and adipose stromal cells. Therefore, the International Federation of Adipose Therapeutics (IFATS) adopted the ADSCs nomenclature for more uniformity.

These cells are currently isolated from the subcutaneous AT [63], which allows for them to be rapidly acquired in large numbers and with a high cellular activity [1,2,64]. Their frequency harbors nearly 2% in its stromal vascular fraction and it is considered to be the greatest one in all tissues [65,66], sometimes nearing 30% [67]. The scientific consensus, the IFATS, and the ISCT agree on the lower expression level of stromal associated markers CD13, CD29, CD44, CD73, CD90, CD105, and CD106 in the SVF population on the contrary to cultured ADSCs. These expressions were consistently pronounced by late ADSCs culture passage [47,57,68]. The specific expressions of CD10, CD36, and CD106 can distinguish them from those of BM [32,69]. ADSCs research, being predominantly carried out using culture-expanded cells, has rather led to a recent acceptance of CD34 as a marker for isolated ADSCs. Thus, there remain interesting aspects of CD34 biology to be explored and understood. A relationship between this CD34 marker and hypoxia has also been reported. CD34 might be a niche-specific marker of progenitors, and hypoxia is related to the maintenance of adult stem cells [70]. Sengenès et al. has demonstrated that the human SVF cell populations presenting the CD34 were enriched in ADSCs, on the contrary to BM-MSCs [71]. However, the expression of this antigen was only reported in the early culture passage [72]. In addition, the CD34+ subpopulation represented at least 20% of freshly isolated SVF [73]. Another surface antigen Stro-1, which is the classic BM-MSCs associated antigen, was differently reported on ADSCs within the literature [2,74]. Furthermore, one report has indicated that ADSCs might be specifically identified by the CD271 marker [75], where the expression have been maintained in elderly people and associated with high proliferative and differentiation abilities [76]. ADSCs also secrete trophic factors that regulate cell growth and display lipolytic responses to β-adrenergic agents and activated protein kinase phosphorylation in tumor necrosis [77,78].

ADSCs from younger donors exhibited a higher proliferation rate when compared to elderly ones, but the differentiation capacity was maintained with aging [79], thus having advantages on BM-MSCs. ADSCs also maintained their potential to differentiate into cells of mesodermal origin and they are commonly known for their low immunogenicity and modulatory effects [33,80]. Less than 1% of them expressed the HLADR protein on their surface, leading to immunosuppressive effects and making them suitable for clinical applications in allogeneic transplantation and in therapies for the treatment of resistant immune disorders [37,66]. Even UC-MSCs were presented as more primitive than ADSCs [5], with the latter have proven their superiority regarding availability and suitable cell quantities. As for BM, ADSCs cell growth stopped in passage 11–12, but presented the lower population doubling than BM and UCB-MSCs, hence being shorter but more highly proliferative than UC-MSCs [5]. These ADSCs features altogether seemed to promote them in tissue repair, where cell proliferation, angiogenesis, and anti-inflammatory processes were expected to occur rapidly in damaged sites (Figure 1).

From another point of view, the ADSCs were firstly identified by their ability to differentiate into mesodermic lineages. However, many findings agreed on latter, with their differentiation into ectodermic and endodermic germinal layers exhibiting a potential advantage in developing tissue engineering strategies in regenerative medicine. Even ADSCs were more susceptible to differentiate into the adipogenic lineage when compared to BM- and UC-MSCs, their multipotency was appreciated for ectodermic and endodermic tissue repair [6,30,47,81] (Figure 2). Nevertheless, their functionality was impacted by long term expansion culture and cryopreservation [32,64,82], which suggests that further studies should be performed to standardize ADSCs manipulation for clinical use.

## 4. Current Approaches in Adipose Derived Stem Cells (ADSCs)-Based Therapeutic Applications

Pluripotent or resident stem cells participle and induce all of the biological mechanisms that are involved in normal process of tissue regeneration and functionality during the whole life. However, MSCs seemed to be more adaptable to act in situ, thus leading to locally support degenerative diseases.

ADSCs are considered to be tools for replacing, repairing, and regenerating dead or damaged cells. These cells were included in clinical investigations that belong to therapeutic strategies and fulfill the general accepted criteria for cell-based therapies, making them the most adapted cell to play the primary role in regenerative medicine. Indeed, ADSCs are found in abundant quantities and they are harvested by a minimally invasive procedure, can differentiate into multiple cell lineages in a regulatory and reproducible manner and they are safely transplanted at the both autologous and allogeneic setting. Their possible manufacturing in accordance with current Good Manufacturing Practices (cGMP) guidelines is in progress by resolving different technical difficulties.

Clinical studies generally used the whole SVF, isolated ADSCs (alone or in combination with biomaterials), even being autologous or allogeneic. Table 1 summarizes the published results of the most reported clinical applications of ADSCs in parallel to those of BM-and UC-MSCs. To avoid misunderstandings on the use of SVF and its derived ADSCs population and their relative beneficial characteristics, it is important to distinguish between these two populations and, hence, only ADSCs applications are viewed in this table. Previous reviews have already reported the performing clinical applications of autografts that are enriched with ADSCs and or SVF and their relative benefit use in regenerative medicine [83].

Studies in Table 1 altogether reported significant differences in transplanted cells, the route of their administration and their clinical status. ADSCs represented the most benefit-to-risk in skin, onco-hematology, bone, and cartilage applications when compared to BM- and UC-MSCs. The use of these cells was related on multiple purposes, including immunomodulation effects (multiple sclerosis MS, fistula, acute and chronic GVHD, diabetes mellitus I, Crohn’s diseases), angiogenic potential (ischemia, scars, wound repair), and differentiation potential (arthritis, cardiac and spinal injury, bone regeneration). The delivery method of stem cell implantation is an important determinant for effective cell therapy.

### 4.1. The Effects of ADSCs on Wound Healing and Skin Regeneration

The stem cell-based therapies in wound healing were firstly performed with autologous cultured epidermal autografts (CEA) as an epidermal substitute. The addition of cellular components to these skin substitutes have led to a satisfying skin esthetic appearance and function. The use of MSCs from different sources has emerged as a unique alternative for wound management strategies to various skin-related disorders to have a near-natural skin in terms of appearance, texture, color, and metabolic properties. Regarding their accessibility, high cell number availability, and their non-invasive collection, ADSCs have been proposed to overcome CEA limitations as recurrent open wounds, long term fragility, and increased scar contractions by inducing extensive ECM protein production [7]. Their ability to restore chronic wounds was increasingly highlighted through ECM secretion, leading to proliferation and remodeling phases of wound healing [8]. These cells have been shown to promote fat tissue survival and, together with free fat, were becoming a real alternative to soft tissue augmentation surgery, as in breast augmentation and facial tissue defects [10,11,12,13] (Table 1).

ADSCs induced tissue reconstitution, thus correcting tissue defects and improving skin regeneration, helped cicatrization and modulated inflammation, thereby promoting skin healing [16,17]. The growth factors and anti-inflammatory cytokines that they secrete prevented cell apoptosis and induced neo-angiogenesis, particularly in treating critical limb ischemia [90]. Many groups have used autologous or allogenic ADSCs in the treatment of burns, their use alone, or in combination with epidermal graft have improved skin engraftment.

Some therapeutic advances have been showed when using BM- or UC-MSCs to achieve the complete remission for patients that were treated for ulcers, scars, and burns [84]. However, the ability of ADSCs to act through higher proliferation, differentiation, and paracrine manner has proven their superiority for widespread application in this field.

### 4.2. The Effects of ADSCs in Auto-Immune Disorders

ADSCs have been applied for many autoimmune disease’s treatments. These cells were shown safe and efficient for Systemic Lupus Erythematosus (SLE), systemic sclerosis (SS), Sjörgren’s syndrome, scleroderma, and Crohn’s diseases. Patients with Crohn’s diseases presented closed fistula after inoculation of the cells [141,142,143] (Table 1), and disease improvement, combined with patient’s safety, have been reported with no serious adverse events being estimated for short or long-term follow-up. At the other hand, a decrease in disease severity was observed after BM and UCB-MSCs treatment and this effect appeared to be up-estimated perhaps for the small sample size or the short-term follow-up [132,133,134,136]. However, many of the conducted studies reported incidences of infections and malignancies. These adverse events might not be attributable to cell infusion, but rather to the impaired immune system and the use of immune-suppressers and corticosteroids as autoimmune medications [196]. Nevertheless, these cells were reported to be the most beneficial in treating and preventing Graft-Versus-Host Disease (GVHD) resulting from HSCs transplantation [167].

ADSCs approaches appeared to be safer and more efficient in terms of cases of side effects reported. Designing MSCs transplantation protocols while using ADSCs therapeutic purposes might be predominantly selective for patients with Sjögren’s syndrome, SLE, polymyolitis, and Crohn’ disease [15,143,144] (Figure 3). The immunomodulating potential of ADSCs has also benefitted from type I diabetes mellitus, where new cells secreting insulin were regenerated [145].

### 4.3. The Effects of ADSCs on Hematological Disorders and Graft-Versus-Host Disease (GVHD)

The treatment of GVHD has been the most widely investigated with BM-MSCs after allogeneic HSCs transplantation and or lymphocyte infusion (Table 1). In adult patients and children with steroid-resistant acute GVHD, the infusion of allogeneic BM-MSCs resulted in the complete response and overall survival in most patients after UCB and HSCs transplantation without any side-effects during or immediately after MSCs infusion [167,168], while better outcomes were also observed with early BM-MSCs administration [169] (Figure 3). In malignancies, BM-MSCs were safe and successful in improving hematopoietic engraftment [170,197], and similar results have been reported with UCB-MSCs in pediatric and adult patients after HSC transplantation [175,176]. Accelerating hematological recovery was also found after the co-infusion of autologous blood HSC and BM-MSCs in patients presenting breast cancer that were receiving high-dose chemotherapy [198].

ADSCs, as their counterparts, were also reported to support HSCs engraftment preventing acute and chronic GVHD [14]. These cells were found to support the complete differentiation and production of mature myeloid and B lymphoid cells, but they were unable to maintain long-term survival and self-renewal of HSCs, suggesting that ADSCs could be only efficient in short term reconstitution of hematopoiesis [181]. Similar results were found in supporting the in vitro and in vivo hematopoiesis in mice [199].

### 4.4. The Effects of ADSCs on Bone/Cartilage Repair

Unlike other adult tissues, which generate scar tissue in response to injury, the skeleton undergoes regenerative healing, forming new bone [200]. In recent years, there has been growing interest in the use of undifferentiated progenitor cells for tissue engineering, owing to their ability to expand in culture and to differentiate into multiple cell lineages when cultured under specific growth conditions. Owing to these characteristics, adult stem cells from different tissues have been used in various cartilage and bone regenerations. ADSCs have shown significant chondrogenic potentials for use in tissue engineering approaches [201]. 

The widely accepted use of ADSCs is, unsurprisingly, bone and cartilage regeneration and repair. Although possessing high osteogenic differentiation potency in vitro, BM-MSCs osteogenic repair in vivo remained limited, their use for bone repair is now progressively replaced with ADSCs. As the same way, limited evidences were observed in regards to the clinical benefit of UCB-MSCs for articular cartilage repair and only one clinical trial has been performed [149] (Table 1).

Alone, or combined with scaffold, ADSCs have proven efficiency in restoring maxillary defects, accompanied with apparent bone regeneration [164]. Post-traumatic calvarial defects were also treated with autologous ADSCs and calvarial continuity with bone regeneration were obtained [165]. Their ability to secrete anti-inflammatory factors and decrease pro-inflammatory responses also makes them very attractive and efficient in the treatment of inter-vertebral disc damages, rheumatoid arthritis, osteoarthritis, and tendon damages [19,22,150,151,152,153,154]. The therapeutic benefit of ADSCs was more pronounced in cartilage defects treatment, and more case treatments were reported in the literature [18].

### 4.5. The Effects of ADSCs on Cardiovascular and Muscular Diseases

Adult skeletal muscle possesses extraordinary regeneration capacities. The muscle satellite cells are responsible for the postnatal growth and major regeneration capacity of adult skeletal muscle. The evidence indicates that skeletal muscle influences systemic aging. An age-dependent decline in skeletal muscle mass, strength, and endurance during the aging process is a physiological development and several factors may exacerbate this process. Capillaries are an integral part of the mechanism underlying this close matching between the blood flow and metabolism of skeletal muscle mass. In contrast with the skeletal muscle, limited regenerative capacity characterizes the myocardium. The neonatal mammalian heart can substantially regenerate after injury through cardiomyocyte proliferation until postnatal day 7. One of the factors that was shared by organisms was the ability able to regenerate the heart is the oxygenation state. It has been hypothesized that the transition to the oxygen-rich postnatal environment is the upstream signal, which results in cell-cycle arrest of cardiomyocytes [202].

Cardiac muscle has limited proliferative capacity, and regenerative therapies are in high in demand as a new treatment strategy. Fibroblasts can be directly reprogrammed to cardiomyocytes by overexpressing a combination of three cardiac-specific transcription factors (Gata4, Mef2c, Tbx5) [203]. Interestingly, it was reported that somatic cells can be directly reprogrammed to alternative differentiated fates without first becoming stem/progenitor cells. Evidence that the exposure of human skin fibroblasts to a Radio Electric Asymmetric Conveyer (REAC) provided commitment toward cardiac and skeletal muscle lineages [204].

Efficient methods to induce differentiation to cardiomyocytes that generate homogenous populations of cardiomyocytes of sufficient quality are a prerequisite for new applications. MSCs can be isolated from various human tissues, with multipotent and immunomodulatory characteristics to help in damaged tissue repair. Clinical trials have demonstrated the feasibility and efficiency for CVD therapy from diverse origins and tissue-derived MSCs [205]. In the field of the new strategies of obtaining cardiac differentiation, stem cell-based therapy has emerged as a promising therapeutic approach for treating cardiovascular diseases [206]. Evidence suggests that secreting paracrine factors prominently mediate the therapeutic effects of transplanted stem cells, importantly, microRNAs (miRNAs) are present in the secreted exosomes [207]. MSC-derived exosomal miRNAs exert cardioprotective effects through the induction of angiogenesis in ischemic heart [208].

Numerous studies employing small animals have adopted intramyocardial injections. However, these injections still show the limitation of cellular engraftment in the infarcted myocardium. Many attempts have been proposed to overcome this obstacle, like a few novel hydrogels have been introduced as potential delivery methods for ADSCs. MSCs were reliably accepted to differentiate into the cardiomyocytes in vitro [209] and they hold great promises in regenerating cardiomyocytes in the cases of myocardial infarction. Autologous and allogenic BM-MSCs that were transplanted into patients with myocardial infarcts showed no adverse side effects and increased the left ventricular function, decreased cardiac arrhythmias, and improved myocardial function [183,184,185] (Table 1). Other studies used UCB-MSCs as intravenous infusion into patients with heart failure, reduced ejection fraction, and improvement of the left ventricular function were attained in these patients [187,188].

Recently, direct cardiac reprogramming has emerged as a novel technology to regenerate damaged myocardium by directly converting endogenous cardiac fibroblasts into induced cardiomyocyte-like cells to restore cardiac function. Fibroblasts replace dead cardiomyocytes, leading to the formation of fibrosis and myocardial remodeling. There are studies regarding the interaction between ADSCs and fibroblasts. ADSCs conditioned media promotes fibroblast proliferation, which suggests the paracrine activation of fibroblasts by ADSCs. The same fibroblasts that were cultured in ADSCs conditioned media were found to secrete increased amounts of type I collagen. These findings suggest that these interactions might play a major role in myocardial protection. 

In another point of view, ADSCs might be for a great cardiovascular interest and these cells were investigated in many clinical trials in last decade. ADSCs were directly administrated into myocardial tissue of patients with MI or ischemic heart failure without related events and their use was safe and efficient in improving cardiac function [191,192,193]. In animal models, ADSCs also have proven efficiency, even when administered by intracoronary, transendocardial, intramyocardial, or intravenous methods [66,210].

In skeletal muscle, the regenerating ability of MSCs is still controversial and few human studies were reported. However, clinical promising reports have shown the improvement of muscle activity in patients presenting muscle dystrophies after the infusion of UC-MSCs [189,190]. However, the paracrine activity of these MSCs was more expected to improve cardiac function by enhancing angiogenesis and anti-apoptosis, rather than direct differentiation towards cardiomyocytes [211,212].

In animal models, ADSCs can be differentiated into skeletal muscle cells and improve the ECM collagen VI deficiency in the congenital muscular dystrophy and in the mice model of Duchene Muscular Dystrophy (mdx mice) [213,214,215,216].

### 4.6. The Effects of ADSC on Neuro-Degenerative Diseases

Conventionally, over a long time, nervous system tissue has been considered to be problematic in regenerating, because mature neural cells do not proliferate or differentiate. It is still unclear how neural stem cells are actively maintained throughout life, and what are the cellular interactions and molecular cascades. In the adult mammalian brain, there are at least three areas that are neurogenic and contain a reservoir of neural stem cells: the subgranular zone in the hippocampal dentate gyrus, the subventricular zone around the lateral ventricles, and the hypothalamus. Recently, several components of the neural stem cell niche have been identified, which regulate neural stem cell activity by supplying various signals [217]. ADSCs could transdifferentiate into neuron-like cells that present multiple neuronal properties, such as synaptic transmission, action potential, secretion of dopamine and neurotrophic factors, and spontaneous postsynaptic current. Some recent studies indicate that miRNAs might regulate the neuronal-like differentiation of MSCs. MiR-21 has been confirmed to directly contribute to ADSCs differentiation [218]. For ADSCs to become ideal for neurological disease therapy, they must generate sufficient number of functional and high-quality neural cells. During trans-differentiation into neural cells, xenobiotics or specific factors and the corresponding partial methylation or acetylation of genomic regions and the activation of further trans-differentiation processes stimulate ADSCs.

These ADSCs and others MSCs have been proposed for novel therapy to various neurological disorders, such as Alzheimer’s and Parkinson’s diseases, stroke, amyotrophic lateral sclerosis (ALS), Huntington’s disease, spinal cord injury, traumatic brain injury, and MS. These, altogether, cells have proven their efficiency to pass through the blood brain barrier even intravenously transplanted. Neuronal morphological characteristics and markers were observed after cell infusion, either by differentiation to neuron-like cells [81] or fusion with endogenous cells [219]. However, their therapeutic benefits appeared to be induced through the secreted proteins [20,21]. Proteomic analysis have confirmed that BM- and UC-MSCs conditioned media induced neuronal differentiation, neurite outgrowth from dorsal root ganglion explants, while ADSCs showed higher axonal growth [20]. Particularly, ADSCs paracrine activity was not solely responsible for the elaboration of nerve growth factors and neurotrophic mediators that are involved in nerve regeneration [21], which adds to the complexity of microenvironment and nerve progenitors’ interactions. Used at autologous or allogenic setting, at low or high infused doses and with multiple infusions, neuroprotective and immunomodulation effects were related to MSCs from the three sources, which leads to a decrease in pathological features and an improvement of the disability of neurological disorders (Table 1).

Therapeutic properties of BM-MSCs have put profit firstly to patients suffering from Parkinson’s disease [220] and these encouraging results pave the way to others current controlled phase study. On another side, safety and clinical improvement have been shown in patients presenting ALS and stroke when using modified BM-MSCs [99,221]. A recent work investigating the BM-MSC-derived neural progenitor on patients with MS has established safety and tolerability without any serious adverse effects [100,135]. Using another alternative, conditioned media from BM-MSCs cultures, followed by BM-MSCs transplantation, potentiated clinical improvement and this ameliorative effect appeared to be dependent on IL-6, IL-8, and vascular endothelial growth factor (VEGF) conditioned the media levels [101]. At the same way, Li et al. has already demonstrated a shift from Th1 to Th2 immunity in human UC-MSCs treated patients with MS [222]. The mainly mechanisms underlying the therapeutic benefit observed for ADSCs are undeniably and irrevocably their paracrine effect after transplantation. The secreted factors might directly act on neuroprotection or through immunomodulation impacting the expression or secretion of many mediators that are involved in angiogenesis, synaptogenesis, gliogenesis, and neurogenesis [223,224,225,226,227]. Other therapeutic strategies might concern cell combination for more neurological function improvement [228] or the use of VEGF, Angiogenin (ANG), and TGF-β as predictive biomarkers for cell therapy effectiveness and for choosing patients [87].

Other ADSCs therapeutic concerns were proposed for retinal degenerative diseases and renal transplantation. These approaches, still in earlier stages of development, were respectively investigated for ADSCs differentiation potency and immunosuppressive ability in tissue repair.

### 4.7. The Effects of ADSCs on Radiation Injuries

The literature is rich in terms of clinical applications of ADSCs to cure radiation injury. There is an increase of radiation injuries on wounds and other organs with the widespread use of radiotherapy, interventional radiological, or cardiological procedures [229,230,231,232,233,234,235,236]. Chronic radiation wounds usually cannot be treated with conventional methods, such as flap surgery or skin grafting because of tissue ischemia and fibrosis [237]. The ischemia is due to inadequate vasculature and incompetent vessels in irradiated tissues [238,239]. The radiated skin shows erythema and abnormal pigmentation. Once a radiation wound is present, it becomes complicated with necrosis, infection, and fibrosis in various organs, such as the heart. These chronic radiation injuries could be improved by sufficient blood supply to the tissues. On the other hand, combined radiotherapy and chemotherapy have represented a major advance in the therapeutic management of cancer therapy. However, the combination of doxorubicin (DXR) and cardiac irradiation could precipitate the unexpected expression of congestive heart failure [240,241].

ADSCs therapy is promising in the treatment of chronic radiation wounds and myocardial diseases. Their administration leads to improved blood perfusion and capillary density in irradiated wounds [235]. The viability of irradiated skin flaps increased when treated with ADSCs injection in correlation with increased vascularity in the flaps injected with ADSCs. At the cellular level, the ADSCs were shown to stimulate fibroblasts proliferation and increase the expression of several cytokines, such as VEGF [236].

## 5. Issues Related to Autologous and Allogeneic Clinical Use of ADSCs

The stemness characteristic, plasticity, and robustness that were attributed to MSCs make them the most attractive adult stem cells in regenerative medicine. As the number, frequency, and differentiation capacity of BM-MSCs negatively correlate with age [35,49], elderly patients could not have clinically efficient autologous stem cells, suggesting that an allogeneic approach would be required. The development of allogeneic approach means that ADSCs will be isolated from a volunteer donor, expanded ex-vivo, and cryopreserved as suitable cell product until need for tissue repair. Thus, cells that were obtained from a single donor can be used to treat thousands of unrelated patients. AT from HLA identical siblings, haplo-identical relatives, or HLA-screened healthy volunteers is now considered to be a successful alternative and it might be the best choice for collection and storage until used in HLA-matched patient. 

Even AT has important implications in the development of stem cell bank [242,243,244], the interest raises by ADSCs have improved two specific questions: first, their use must be practical and effective (high numbers of cells are needed); and second, the clinical outcomes should be identical to the expected use. A major advantage is that of ADSCs could be maintained for up to 24–48 h within lipoaspirates [82], cryopreserved before separation and seeding in culture with a stable and efficient ability to proliferate and differentiate [245,246], while fresh BM and UC tissues are necessary for the collection and deriving of MSCs. Consequently, the banking of allogeneic ADSCs remains full of hope for the future regenerative medicine. However, to be used on a widespread basis, efficient, simple, and especially safe methods should be provided to ensure the assurance of available cell quantities of no contaminated and functional ADSCs and it performed accordingly to cGMP. Specific insights are now focused on how these procedures should be realized with regards to their therapeutic benefits [78,247].

Many studies have been focused on this field in different diseases, but the availability of higher quantities of qualified ADSCs for banking is still debated due to the lack of best and standardized parameters of cryopreserved cells. Hence, the guidance policies covering all the processes involved in ADSCs banking (donor’s recruitment, manipulation, banking procedures, release, and defined qualification testing after thawing) would guarantee multiple ADSCs patient treatments avoiding repeated liposuctions. 

We can also speculate that the allogenic model would be more attractive for cell therapy industry. However, autologous cells, rather than allogenic ones, mostly improved cell-based immunotherapy.

Despite the major interest arising concerning their use in therapy, the last decade still lacked commercial GMP-licensed products. The proposed ADSCs products appeared to be well tolerated in clinical phase I studies, provided the dosage data in phase II, and some of them were in phase III, such as in cardiovascular and autoimmune diseases. Besides academic and university communities, some biotechnological firms are now covering clinical ADSCs research for many areas. Developed laboratory ADSCs preparations or patented formulations of adult stem cells are available to undergo and evaluate the safety of different MSCs treatment for many diseases. Osiris Therapeutics Inc., Mesoblast Limited is committed to develop stem cell-based treatment to Crohn’s diseases, cardiovascular diseases, diabetes, orthopedic disorders, and radiation exposure [248]. These human BM expanded MSCs, called “Prochymal”^®^, appeared to be safe and efficient and improve the treatment of acute GVHD [171]. However, a major concern giving rise to ethical and justice considerations is the cost of these biotechnological products. ADSCs and derivatives might also be costly and labor-intensive, thus increasing the cost of making them available. The way that clinical trials are conducted should aim to reduce the unfair disparities in access.

## 6. Summary and Conclusion

ADSCs represents many therapeutic challenges in terms of origin, type, and the manner to use them, different recent investigations pave the way to their successful therapeutic use in tissue repair. More insights into standardizing technical use are warranted to evaluate the in-depth efficacy and safety of ADSCs-based therapy and evaluate the benefit-to-risk ratio in clinical applications. The beneficial effects of stem cells, and there with the paradigm of tissue regeneration, may not be restricted to cellular restoration, but may also be related to the transient paracrine actions of the cells.

These cells are phenotypically identical to BM- and UC-MSCs when using a panel of 22 surface antigens [35], the gene expression signature seems to demonstrate an up-regulation of 24 genes in ADSCs when compared with BM-MSCs [50], and several hundred expressed sequence tags [35]. The variations that were described between ADSCs and BM-MSCs regarding differentiation potential and proliferation capability may also reflect the incidence of the micro-environment of their tissue of origin. This fact might answer the differences reported in their ex-vivo expansion and promising clinical benefit. Indeed, the promising result that was observed when transplanting SVF might be refereed to its heterogeneity and the activation of the related microenvironment that was constituted by subpopulations of hematopoietic and no hematopoietic and endothelial progenitors and cells. Hence, insights into host-related factors, including local environment and optimal timing are needed to master factors governing the fate of ADSCs after infusion. Patient’s associated factors should also be considered in designing ADSCs based therapy, due to their influence on the number and functional behavior of expanded cells [249,250].

We should emphasize ADSCs purity control and identify potency assays to ensure their regenerative potency. These factors imply the identification of reproducible and consistent standardized cell preparations that are suitable for use in a large scale. Actual application may need more expanded numbers of cells and variabilities reported in clinical outcomes might be related to differences in the infused cell amounts.

All these considerations would act to reinforce their promising therapeutic benefit for tissue repair. As expected by Gimble et al., may be one day people will altruistically participate in “fat drives.” for fat donation as observed in “blood drives” [251]. Clinical grade ADSCs applications should be performed in compliance with appropriate standards and protocols relative to cellular therapies from facility and GMP compatible reagents to the clinical grade materials. Therapeutic development of HSC transplantation witnesses the success of the global standardized processes and the generalization of guidelines regarding the clinical application of these cells. ADSCs-based therapies could shortly follow the example of HSC’s therapy process.

The application of ADSCs is greater than that of BM-MSCs in regenerative medicine, because ADSCs have the need for easier technique for isolation when compared to BM-MSCs, because the technique for isolating ADSCs is easier and, consequently, they can be used in a greater number than BMSCs. The clinical use of cell-based therapies in regenerative medicine has to fulfill the minimal requirements, like any other medical treatment. Still, before ADSCs can be applied for routine clinical applications, many open questions that are related to ADSCs need to be solved. Several reports have suggested that ADSCs promote the proliferation of cancer cells that means ADSCs may stimulate the growth of pre-existing tumors. Consequently, the treatment with ADSCs in cancer diseases is not presently recommended 

In conclusion, considerations should be made regarding the scientific and medical challenges for the requirement of widely use of banked ADSCs. For the medical ones, health care professionals need to be educated for the correct use and applications of ADSCs. At the other scientific side, challenges should improve the appropriate quality assurance and control in compliance with cGMP.

## Figures and Tables

**Figure 1 ijms-20-02523-f001:**
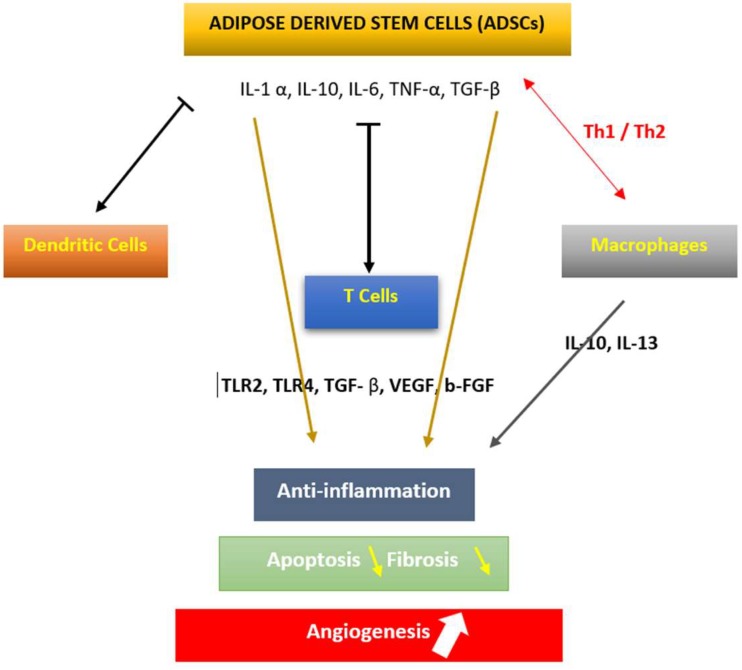
Immuno-modulatory effects of adipose derived stem cells (ADSCs). ADSCs stimulate macrophage change immunity and inhibit T and Dendritic cells, inducing angiogenesis, a decrease in apoptosis and fibrosis with an increase in anti-inflammation process. Interleukin-1α, -6, -10 (IL-1α, -6, -10, -13), TNF-α (Tumor Necrosis Factor-α), TGF-β (Tumor Growth Factor-β), TLR2, TLR4 (Toll Like Receptor 2, 4), VEGF (Vascular Endothelial Growth Factor), b-FGF (basic Fibroblast Growth Factor).

**Figure 2 ijms-20-02523-f002:**
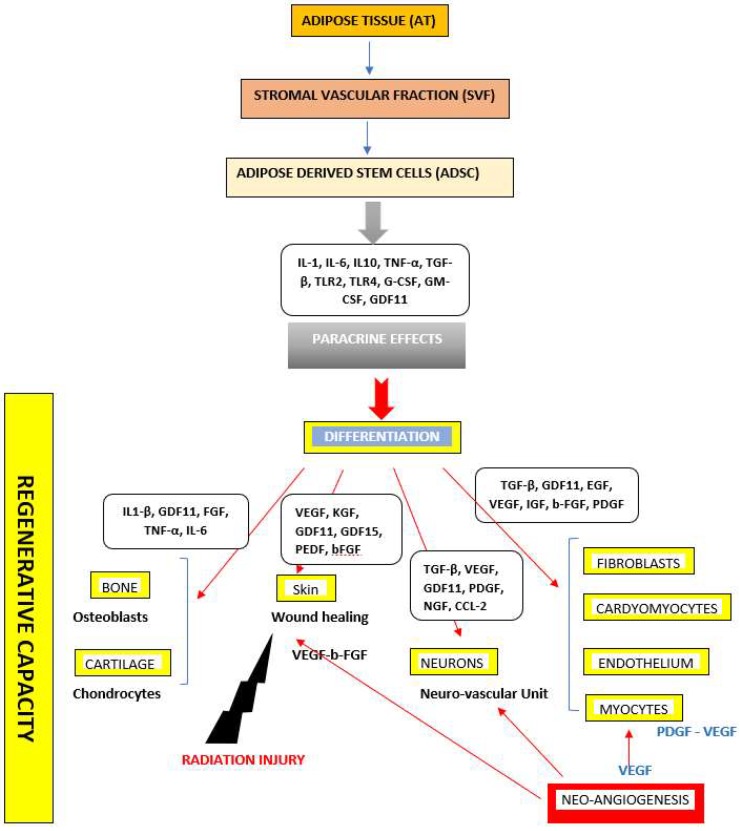
Secretome of Adipose Derived Stem Cells (ADSCs) that are involved in the mechanisms related to tissue repair and regeneration. ADSCs secrete different growth factors in their microenvironment and other proteins known to induce specific cell differentiation. Interleukin-1, -6, -10 (IL-1, -6, -10), TNF-α (Tumor Necrosis Factor-α), TGF-β (Tumor Growth Factor-β), TLR2, TLR4 (Toll Like Receptor 2, 4), GDF11 (Growth Differentiation Factor 11), GDF15 (Growth Differentiation Factor 15), G-CSF (Granulocyte-Colony Stimulating Factor), GM-CSF (Granulocyte Monocyte-Colony Stimulating Factor), EGF (Endothelial Growth Factor), VEGF (Vascular Endothelial Growth Factor), IGF (Insulin Growth Factor), b-FGF (basic Fibroblast Growth Factor), PDGF (Platelet Derived Growth Factor), NGF (Nerve Growth Factor), CCL-2 (Chemokine C-C Motif Ligand 2), PEDF (Pigment Epithelium Derived Factor), KGF (Keratinocyte Growth Factor).

**Figure 3 ijms-20-02523-f003:**
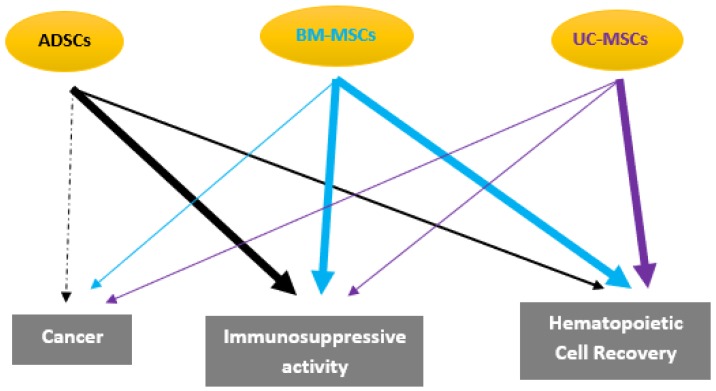
Therapeutic features of Adipose Derived Stem Cells (ADSCs) in auto-immunes and hematopoietic diseases compared to those of bone marrow (BM)- and umbilical cord (UC)-mesenchymal stem cells (MSCs). The size line is related to the observed effect.

**Table 1 ijms-20-02523-t001:** Summary of results of previous clinical studies using ADSCs as compared to those with BM-and UC-MSCs. BM: bone marrow, UC: umbilical cord, ADSCs: adipose derived stem cells, MSCs: mesenchymal stem cells, Auto: autologous, Allo: allogenic, AD: dermatitis, ALS: amyotrophic lateral sclerosis, MS: multiple sclerosis, SCA: spinocerebellar ataxia, SLE: systemic lupus erythematosus, LN: lupus nephritis, SS: systemic sclerosis, GVHD: graft versus host disease, BMD: Becker muscular disease, DMD, Duchene muscular disease.

Stem Cell-Based Therapy	References	Cell Origin	Auto/Allo	Associated Effects	Application
**Skin**	[84,85,86]	BM-MSCs	Auto	Modulation of inflammation, wound repair	Radiation burns, burns
	[87,88,89]]	UC-MSCs	Allo	Wound healing, skin improvement, neoangiogenesis and ulcer healing, decrease in features associated with AD	Burns, wound healing, severe diabetic foot, Atopic Dermatitis (AD)
	[10,11,12,16,17,90,91,92,93,94,95,96,97,98]	ADSCs	Auto	Wound healing, traumatic scars, facial tissue defects, post-mastectomy radiation, breast augmentation, facial rejuvenation, hair loss and growth, breast reconstitution, neogenesis across affected arteries	Soft tissue augmentation, esthetic remodeling, anti-aging, ulcers, burns, alopecia, lipodystrophies, critical limb ischemia
**Nerve system**	[99,100,101,102,103,104,105,106,107,108,109,110,111,112,113,114,115,116,117,118,119]	BM-MSCs	Auto Allo	Safety and efficacy, increased functional recovery, improvement, slowdown of Amyotrophic Lateral Sclerosis (ALS) progression, safety in ALS, spinal cord repair, expanded disability scale score improvement, improvement treatment of MS	Ischemic stroke, traumatic brain injury, spinal cord injury, ALS, chronic spinal cord lesions, multiple sclerosis (MS), Parkinson, spastic cerebral palsy, refractory epilepsy, chronic complete paraplegia
	[5,120,121,122,123,124,125]	UC-MSCs	Allo	Recovery of neurologic function, neuroprotection from sclerosis, delay SCA progression, safety and effectiveness, feasible and safety approach in stroke, self-care in patients	ALS, hereditary spinocerebellar ataxia (SCA), MS, Stroke in Middle Cerebral Artery, Traumatic Brain Injury sequelae
	[126,127,128,129,130,131]	ADSCs	Auto	Recovery from ischemia, safety	Spinal cord injury, ALS, MS
**Autoimmune disorders**	[132,133,134,135]	BM-MSCs	Auto	improvement the clinical condition, reduction of Crohn’s disease and perianal diseases activity	Crohn’s diseases, perianal diseases, SLE
	[112,132,136,137,138,139,140]	UCB-MSCs	Allo Allo	Safety and improvement of insulin secretion after HSC transplantation, renal remission for LN patients, safety and increase T cell level, partial disease remission, Safety with adverse events	Diabete I, Diabete II, Lupus Nephritis (LN), Systemic Lupus Erythematosus (SLE), systemic sclerosis (SS), Sjörgren’s syndrome
	[15,141,142,143,144,145,146,147,148]	ADSCs	Auto Allo	Proinflammatory cytokine decrease, improvement safety and efficacy treatment, Generation of Insulin-secreting cells, improvement in disease activity and safety	perianal and rectovaginal fistules, Crohn’s diseases, Type 1 Diabete Mellitus, refractory SLE Polymyolitis, SS
**Cartilage**	[63,117]	BM-MSCs	Auto	Good integration in bone	knee chondral lesions, cartilage defects
	[149]	UC-MSCs	Allo	Cartilage regeneration	Osteoarthritis
	[19,22,150,151,152,153,154,155,156,157,158]	ADSCs	Allo Auto	Tendon fibers arrangement, suppressive activity and decrease inflammatory responses, Safety, Proinflammatory cytokine decrease, improvement of knee joint	Intervertebral disc damage, Rheumatoid arthritis disease, Osteoarthritis
**Bone**	[159,160,161]	BM-MSCs	Auto	Large bone diaphysis, good scaffold integration with host bone	Durable bone regeneration, bone defects, bone segment loss, bone diaphysis defects
	[162,163]	UCB-MSCs	Allo	Decreased in healing time in bone callus formation and marrow flow	Bone nonunion, Necrosis of Femoral Heads
	[164,165,166]	ADSCs	Auto Allo	Bone regeneration, calvarial continuity	Maxillary reconstitution, traumatic calvarial defects
**Immune and Hematological Disorders**	[167,168,169,170,171,172,173,174]	BM-MSCs	Allo, Auto	Improvement of HSC engraftment, Lymphocyte recovery, Hematopoietic recovery, Safety and no adverse events	Acute and severe GVHD, Hematopoietic malignancy
	[175,176,177,178,179,180]	UC-MSCs	Allo	Neutrophil and platelet engraftment, successful engraftment, Safety, Reduced graft failure, Decrease in GVHD symptoms	Acute leukemia, Hematologic malignancy, GVHD, Steroid-resistant severe and acute GVHD
	[14,181,182]	ADSCs	Auto,	Improvement of HSC engraftment, immunosuppressive activity, short term hematopoiesis, Renal transplantation immuno-suppressive minimization	Acute and chronic GVHD, Hematopoiesis support, engraftment
**Cardio-vascular and Muscle**	[183,184,185,186]	BM-MSCs	Auto, Allo	improved left ventricular function, decreased cardiac arrhythmias, improvement myocardial function, Decrease in infarct size	Heart infarct, Ischemic cardiomyopathy
	[187,188,189,190]	UCB-MSCs	Allo	Improvement of muscle size and activity, stabilize muscle power in DMD, decrease in infarct size, improve left ventricular function	Becker muscular disease (BMD), Duchene Muscular Disease (DMD), Coronary chronic total occlusion
	[191,192,193,194,195]	ADSCs	Auto Allo	Reduction in myocardial scar formation, improvement of cardiac function, safe and efficient	Myocardial infarction, ischemic heart disease, Heart failure

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
