# Peer review of "Regenerative Capacity of Adipose Derived Stem Cells (ADSCs), Comparison with Mesenchymal Stem Cells (MSCs)"

_ijms, 2019, doi:10.3390/ijms20102523_

Round 1
Reviewer 1 Report
Dear Editor,
I send you my comments on the review “Regenerative Capacity of Adipose Derived Stem Cells (ADSC)” (Loubna Mazini et al).
In this review the authors described the actual state of art on regenerative ability of Adipose Derived Stem reporting studies about the ADSC-based cell therapy and briefly some future applications. Although this is not an innovative topic and it was deepened in other works, the manuscript offers an overview about the growing field of ADSC research for their clinical use.
The structure and layout of manuscript was written in an organized way, easy to read and understand, however sometime there were so deep details that can be treat less in details using references. The references used in the manuscript were adequate to support the concepts.
In page 3, in the paragraph “Adipose derived Stem Cells (ADSCs)”, line 130, Authors report the sentence “These cells are currently isolated from the subcutaneous AT…”. Is known that ADSC are easily collected from the lipoaspirate too. In order to improve the understanding of the manuscript, I suggest to report this information including a reference, such as the work of Palumbo P J Cell Physiol. 2015 (“In vitro evaluation of different methods of handling human liposuction aspirate and their effect on adipocytes and adipose derived stem cells.”)
Spelling errors were detected.
In conclusion I consider that the manuscript can be accepted for publication in IJMS.
Kind regards
Author Response
Dear Reviewer,
Please, find attached the new version of the manuscript.
- Some details have been removed in the text, were written in red and blocked, please see line 275-278, line 340-341, line 373-375 and line 376-379.
- Line 130: the article of Palumbo P et al 2015 was added in red, please see line 141.
- Errors have been corrected.
My best regards

Reviewer 2 Report
In the current study, authors explored the Regenerative Capacity of Adipose Derived Stem Cells. They have the analyzed the therapeutic advancement of ADSC in comparison to bone marrow and umbilical cord-mesenchymal stem cells. However, I have following concerns related to this study:
1. P. 2, line no. 68-69, Introduction:
“MSC have been isolated from almost tissues including BM, UC tissue and blood, AT, dentalpulp, synovial fluid, skin, …”. What does….. indicate? Authors should have clearly described or written etc.
2. P. 2, According to subheading 2 (Mesenchymal stem cells and adipose derived stem cells characteristics), authors have not even once mentioned about ADSC. Instead, the title should have been replaced with only mesenchymal cells and later talked about ADSC in next subtitle no. 3.
3. After discussing ADSC, BM-MSC and UC-MSC, a section on their comparative study and advantage of ADSC over others should be added.
4. Authors have included only one figure (Fig. 1), the title of which should have been like ‘Secretome of ADSC involved in tissue regeneration and repair. Furthermore, a comparative pictorial description of ADSC, BM-MSC and UC-MSC should have been included. In addition, more relevant figures should be added to this review to enrich the scientific concepts.
5. In the abstract, authors have stated that their analysis focused on “ADSC rather than whole stromal vascular fraction”. On contrary, in Fig. 1 legend, it has been mentioned that “ADSC within SVF are able to secrete different growth factors”. I think no need of writing “within SVF”.
6. P.4, line no. 166-168: the sentence “When in human body pluripotent or resident stem cells assure tissue regeneration and functionality for the whole life, MSC might offer in situ long-life treatment especially in degenerative diseases.” is incomprehensible. It should be re-written or well-explained.
7. P. 5, the subtitle should be “The effects of ADSC on wound healing and skin regeneration” and “in dermatology” should be deleted.
8. P. 4, line no. 139-140: the words “interesting aspect” have been repeated.
9. P. 11, the subtitle should be “issues related to”.
10. The reference in table should be in last column.
11. The table legend 1 should be revised like “Summary of results of previous clinical studies….”
12. P. 12, the use of “ But we could not forget” is unscientific.
Based on my overall concerns, this review needs extensive revision for consideration of publication.
Author Response
Dear Reviewer,
Please find attached the new version of the manuscript with responses to your comments.
1. P. 2, line no. 68-69, Introduction: please see line 69-74 written in red;
2. P. 2, Subtitle of subheading 2: Mesenchymal stem cells and adipose derived stem cells characteristics was replaced by (Mesenchymal stem cells characteristics), please see line 68;
3. comparison of ADSCs, BM- and UC-MSCs was discussed in red in line 163, line 168-169, line 171-181;
4. Title of Figure 1 in the first manuscript (and becoming figure 2 in the revised version) has been replaced with "Secretome of Adipose Derived Stem Cells (ADSCs) involved in the mechanisms related to tissue repair and regeneration". see line 545;
- A new figure 1 was added, please see 534 and integrated within the text, line 173;
- Another figure 3 was also added, please see line 559 and integrated in the text line 255, line 263;
5. In Fig. 1 legend, within SVF has been removed, please see line 545;
6. P.4, line no. 166-168: was rewritten, please see line 189-191;
7. P. 5,: "in dermatology" has been removed from the subtitle of "The effects of ADSC on wound healing and skin regeneration", please see line 216;
8. P. 4, line no. 139-140: the words “interesting aspect” have been removed, line 150-151;
9. P. 11, the subtitle should be “issues related to”, it has been correctd, please see line 443;
10. The reference in table should be in last column. According to the instruction anthors of the journal, references should be reported in the first column;
11. The table legend 1 has been rewritten in red as "Summary of results of previous clinical studies using ADSCs compared to those with BM-and UC-MSCs", please see line 437-438
12. P. 12, the use of “ But we could not forget” was replaced by But, please see line 472.
My best regards

Reviewer 3 Report
This Review focuses on the use of adipose derived stem cells (ADSCs) for clinical applications. The Authors analyzed the ADSCs properties compared to Umbilical cord MSCs and hematopoietic MSCs and extensively described their application for the treatment of several diseases, as neurological, cardiovascular and auto-immune pathologies. The review is very interesting and well written and suitable for publication after some minor revision.
All the abbreviations (ADSC, MSC, HSC) should be indicated as plural in the whole text: ADSCs exc.
The manuscript is well written but the introduction section, the paragraph line 275, line 379 and paragraph line 403 are very poor in references. Paragraph Line 109 authors could also discuss the gender differences related to Wharton’s jelly derived stem cells and their differentiation capability, doi: 10.1016/j.ejogrb.2018.12.028, see also doi: 10.3390/ijms19051503 for in vivo application; The paragraph on cardiovascular diseases could be implemented, see for example article doi: 10.3390/ijms19092629, and doi: 10.3727/096368912X657297 for direct fibroblast reprogramming. Within the same paragraph strategies to obtain cardiac differentiation could be discussed, see for example doi: 10.2147/DDDT.S44706. eCollection 2013, and doi: 10.3390/ijms20040982, and 10.1016/j.atherosclerosis.2019.03.016.
There are few grammatical and punctuation mistakes, please correct them (as for example line 23, line 142…)
Line 111 please define CB
Author Response
Deear Reviewer,
Please, find attached the new version of the manuscript with responses to your comments.
- introduction was enriched in references in red within the text, please see line 40;
- paragraph line 275: was enriched innew references, please see line 311-327;
- paragraph line 379 was enriched in references written in red, please see line 412-421;
- paragraph line 403 : new references were added in red, please see line 454-464;
Paragraph Line 109: was rewritten in red according to the articles suggested, please see line 122-125 and line 131-135;
- The paragraph on cardiovascular diseases enriched and discussed with the suggested articles, text in red, please see line 311-327;
- Ponctuations and grammar errors have been corrected.
- Line 111 : CB has been replaced by cord blood in red, lease see line 117.
With my best
